# The shape of the bacterial ribosome exit tunnel affects cotranslational protein folding

**Renuka Kudva[1], Pengfei Tian[2], Fátima Pardo-Avila[3], Marta Carroni[1,4], Robert B Best[2], Harris D Bernstein[5], Gunnar von Heijne[1,4]***

[1]Department of Biochemistry and Biophysics, Stockholm University, Stockholm, Sweden; [2]Laboratory of Chemical Physics, National Institute of Diabetes and Digestive and Kidney Diseases, National Institutes of Health, Bethesda, United States; [3]Department of Structural Biology, Stanford University, Stanford, United States; [4]Science for Life Laboratory, Stockholm University, Solna, Sweden; [5]Genetics and Biochemistry Branch, National Institute of Diabetes and Digestive and Kidney Diseases, National Institutes of Health, Bethesda, United States

**Abstract** The *E. coli* ribosome exit tunnel can accommodate small folded proteins, while larger ones fold outside. It remains unclear, however, to what extent the geometry of the tunnel influences protein folding. Here, using *E. coli* ribosomes with deletions in loops in proteins uL23 and uL24 that protrude into the tunnel, we investigate how tunnel geometry determines where proteins of different sizes fold. We find that a 29-residue zinc-finger domain normally folding close to the uL23 loop folds deeper in the tunnel in uL23 Δloop ribosomes, while two ~ 100 residue proteins normally folding close to the uL24 loop near the tunnel exit port fold at deeper locations in uL24 Δloop ribosomes, in good agreement with results obtained by coarse-grained molecular dynamics simulations. This supports the idea that cotranslational folding commences once a protein domain reaches a location in the exit tunnel where there is sufficient space to house the folded structure.
DOI: https://doi.org/10.7554/eLife.36326.001

***For correspondence:**
gunnar@dbb.su.se

**Competing interests:** The authors declare that no competing interests exist.

## Introduction

A large fraction of cellular proteins likely start to fold cotranslationally in the ~100 Å long exit tunnel in the ribosomal large subunit (*Milligan and Unwin, 1986*; *Ban et al., 2000*; *Nissen et al., 2000*), before they emerge into the cytosolic environment. In *E. coli* ribosomes, portions of the 23S rRNA and a few universally conserved proteins line the exit tunnel, *Figure 1A*. The tunnel proteins uL4, uL22, and uL23 consist of globular domains that are buried within the rRNA, and β-hairpin loops that protrude into the tunnel (*Klein et al., 2004*). These loops help stabilize the tertiary structure of 23S rRNA (*Lawrence et al., 2016*) and contribute towards the unique geometry of the tunnel (*Ban et al., 2000*; *Nissen et al., 2000*; *Harms et al., 2001*). uL24 and uL29 are located near the end of the tunnel, and a hairpin loop in uL24 forms a finger-like structure that partially obstructs the tunnel exit port.

Inspired by observation that protein domains fold in different parts of the exit tunnel depending on their molecular weight (*O'Brien et al., 2010*; *O'Brien et al., 2011*; *Trovato and O'Brien, 2016*; *Samelson et al., 2018*; *Farías-Rico et al., 2018*), we now ask what role the geometry of the exit tunnel plays in determining where these domains fold. To explore this question, we employ the same arrest peptide-based approach (and coarse-grained MD simulations) used in our previous studies of cotranslational protein folding (*Nilsson et al., 2015*; *Nilsson et al., 2017*), but with ribosomes that

carry deletions in either the uL23 or the uL24 hairpin loop. Our findings provide strong evidence that the tunnel geometry determines where in the tunnel a protein starts to fold.

## Results and discussion

### The folding assay

Our experimental set-up, *Figure 1B*, exploits the ability of the SecM translational arrest peptide (AP) (*Nakatogawa and Ito, 2001*) to act as a force sensor (*Ismail et al., 2015*; *Ismail et al., 2012*; *Goldman et al., 2015*), making it possible to detect the folding of protein domains in the exit tunnel (*Nilsson et al., 2015*; *Goldman et al., 2015*). In brief, the domain to be studied is cloned, via a linker, to the AP, $L$ residues away from its C-terminal proline. The AP is followed by a C-terminal tail, to ensure that arrested (*A*) nascent chains can be cleanly separated from full-length (*FL*) chains by SDS-PAGE. Constructs with different $L$ are translated in the PURE in vitro translation system (*Shimizu et al., 2005*), and the fraction full-length protein ($f_{FL}$) is determined for each $L$. For linkers that, when stretched, are long enough to allow the protein to reach a part of the exit tunnel where it can fold, force will be exerted on the AP by the folding protein, reducing stalling and increasing $f_{FL}$. (*Tian et al., 2018*), *Figure 1C*. A plot of $f_{FL}$ vs. $L$ thus shows where in the exit tunnel a protein starts to fold and at which linker length folding no longer causes increased tension in the nascent chain.

A number of earlier studies have provided strong support for the notion that the dominant peak in a $f_{FL}$ profile corresponds to folding into the native state (as opposed to, *e.g.*, non-specific compaction of the nascent chain): (i) folded proteins have been visualized in the exit tunnel by cryo-EM of ribosome-nascent chain complexes at $L$-values corresponding to the dominant $f_{FL}$ peak (*Nilsson et al., 2015*; *Nilsson et al., 2017*; *Tian et al., 2018*), (ii) the dominant $f_{FL}$ peak disappears when proteins that depend on metals or other ligands for folding are translated in the absence of the ligand (*Farías-Rico et al., 2018*; *Nilsson et al., 2015*), (iii) the dominant $f_{FL}$ peak corresponds closely to the tether length at which protein domains become resistant to on-ribosome pulse-proteolysis by thermolysin (*Farías-Rico et al., 2018*) or at which folding can be detected by other techniques such as NMR or FRET (*Kemp et al., 2018*), (iv) the amplitude of the $f_{FL}$ peak correlates with the folding free energy of a domain (*Farías-Rico et al., 2018*).

### uL23 Δloop and uL24 Δloop ribosomes

The *E. coli* strains HDB143 (uL23 Δloop; uL23 residues 65–74 deleted) and HDB144 (uL24 Δloop; uL24 residues 43–57 deleted) have previously been shown to be viable (*Peterson et al., 2010*), as is a strain where uL23 has been replaced by a homologue from spinach chloroplast ribosomes that also lacks the β-hairpin loop (*Bubunenko et al., 1994*; *Bieri et al., 2017*). These strains were used to purify high-salt-washed ribosomes that were used to translate proteins in the commercially available PURExpress Δ-Ribosome kit. Analysis of the purified ribosomes by SDS-PAGE and western blotting demonstrated the expected size differences compared to wildtype for the uL23 Δloop and uL24 Δloop proteins, *Figure 1—figure supplement 1*.

### Cryo-EM structure of uL23 Δloop ribosomes

The loop deleted in the uL24 Δloop ribosomes does not interact with neighboring parts of the ribosome, *Figure 2A*, and hence its removal would not be expected to alter the structure of other parts of the exit tunnel. In contrast, the loop deleted in uL23 Δloop ribosomes is located deep in the exit tunnel, *Figure 2B*, ~40–50 Å from the exit and it is not clear *a priori* whether its removal may cause rearrangements in other tunnel components. For this reason, we determined a cryoEM structure of the uL23 Δloop 70S ribosome at an average resolution of 3.3 Å, *Figure 2C*, *Figure 2—figure supplement 1*, and found that the shape of the tunnel remains unchanged in the uL23 Δloop ribosome when compared with wildtype (WT) *E. coli* ribosomes, except for an increase in volume resulting from the absence of the uL23 loop *Figure 2C–D*. We estimated this increase using the POVME algorithm (*Durrant et al., 2011*; *Durrant et al., 2014*). Compared to WT *E. coli* ribosomes, the tunnel volume increases by 2,064 $\text{Å}^3$ in uL23 Δloop ribosomes, see *Video 1*, about 1/3 of the size of ADR1a (5,880 $\text{Å}^3$) calculated by the same method.

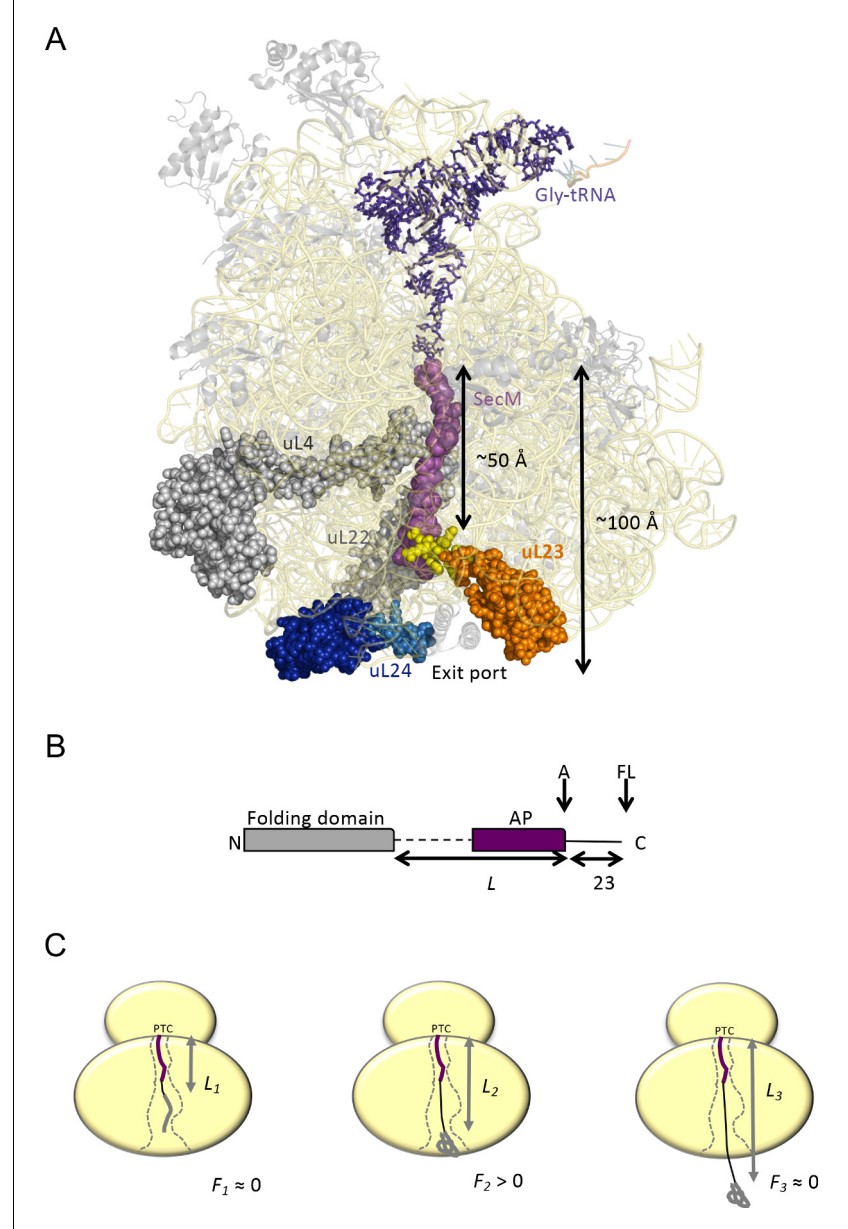

**Figure 1.** Cotranslational protein folding assay. (**A**) Front view of the 50S subunit of the *E.coli* ribosome adapted from PDB 3JBU (*Zhang et al., 2015*), with tunnel proteins uL4 and uL22 indicated in gray. The globular domain of uL23 is indicated in orange with the β-hairpin loop depicted in yellow. uL24 is shown in dark blue, with the loop at the tunnel exit shown in light blue. The exit tunnel, outlined by a stalled SecM nascent chain (purple), is ~100 Å in length. (**B**) The arrest-peptide assay (*Nilsson et al., 2015*). The domain to be studied is placed $L$ residues upstream of the critical proline at the C-terminal end of the 17-residue long arrest peptide (AP) from the *E. coli* SecM protein. A 23-residue long stretch of the *E. coli* LepB protein is attached downstream of the AP, allowing us to separate the arrested (**A**) and full-length (**FL**) products by SDS-PAGE after translation. Constructs are translated in the PURExpress in vitro translation system supplemented with WT, uL23 Δloop, or uL24 Δloop high-salt washed ribosomes for 20 min. The relative amounts of arrested and full-length protein were estimated by quantification of SDS-PAGE gels, and the fraction of full-length protein was calculated as $f_{FL} = I_{FL}/(I_A + I_{FL})$ where $I_A$ and $I_{FL}$ are the intensities of the bands corresponding to the A and FL products. (**c**) $f_{FL}$ is a proxy for the force $F$ that cotranslational folding of a protein domain exerts on the AP. At short linker lengths, both $F$ and $f_{FL} \approx 0$ because the domain is unable to fold due to lack of space in the exit tunnel. At intermediate linker lengths, $F$ and $f_{FL} > 0$ because the domain pulls on the nascent chain as it folds. At longer linker lengths, $F$ and $f_{FL} \approx 0$ because the domain is already folded when the ribosome reaches the end of the AP.

*Figure 1 continued on next page*

*Figure 1 continued*

DOI: https://doi.org/10.7554/eLife.36326.002

The following figure supplements are available for figure 1:

**Figure supplement 1.** $A_{260}$ = 300 units (6.9 µM) of high-salt washed ribosomes were separated on a 12% Bis-Tris gel and transferred by Western blotting onto a nitrocellulose membrane and detected with antibodies against uL24 (panel A) or uL23 (panel B).

DOI: https://doi.org/10.7554/eLife.36326.003

**Figure supplement 2.** Multiple sequence alignment of uL23 and uL24.

DOI: https://doi.org/10.7554/eLife.36326.004

## ADR1a folds deeper in the exit tunnel in uL23 Δloop but not in uL24 Δloop ribosomes

ADR1a constructs of different linker lengths ($L$) were translated in the PURExpress Δ-Ribosome kit supplemented with purified WT, uL23 Δloop, or uL24 Δloop ribosomes, either in the presence of 50 µM $Zn^{2+}$ (to promote folding of ADR1a) or in the presence of 50 µM of the zinc-specific chelating agent TPEN (to prevent folding of ADR1a; TPEN is required to remove residual amounts of $Zn^{2+}$ from the PURE lysate) (*Figure 3A*, *Figure 3—figure supplements 1–4*). Translation rates in PURE are ~10 fold slower than in vivo (*Capece et al., 2015*), but since the proteins studied here fold on micro-to-millisecond time scales, that is considerably faster than the in vivo translation rate, it is safe to assume that the folding reaction has time to equilibrate between each translation step both in vivo and in the PURE system.

Similar to previous results (*Nilsson et al., 2015*), we saw efficient stalling when ADR1a was translated in the presence of TPEN at $L \geq 19$ residues, *Figure 3—figure supplement 8*. Further, there is a slight but significant increase in $f_{FL}$ at $L$ = 17 residues in the presence of TPEN (hence not related to folding); this has been observed before (*Nilsson et al., 2015*) and we hypothesize that it is due to a weakening in the arrest potency of SecM by the ADR1a residues that abut the AP in this construct (see *Figure 3—figure supplement 9* for sequences). To correct for this effect, we calculated $\Delta f_{FL} = f_{FL}(Zn^{2+}) - f_{FL}(TPEN)$, *Figure 3A*. In the presence of $Zn^{2+}$, the $\Delta f_{FL}$ profiles for WT and uL24 Δloop ribosomes are very similar: $\Delta f_{FL}$ starts to increase around $L$ = 20–21 residues and peaks at $L$ = 25 residues (gray and blue curves). In contrast, for the uL23 Δloop ribosomes, $\Delta f_{FL}$ starts to increase already at $L$ = 17 residues and peaks at $L$ = 21–24 residues (red curve). To quantify these differences, for each $f_{FL}$ curve we calculated the linker lengths characterizing the onset and end of the peak ($L_{onset}$ and $L_{end}$; defined as the $L$-values for which the curve has half-maximal height, as indicated in *Figure 3A*), as well as the $L$-value corresponding to the peak of the curve ($L_{max}$), *Table 1*.

A previous cryo-EM study demonstrated that the 29-residue ADR1a domain folds deep inside the ribosome exit tunnel in a location where it is in contact with the uL23 loop (*Nilsson et al., 2015*), *Figure 2B*. The additional space available in uL23 Δloop ribosomes makes it possible for ADR1a to start to fold at 3–4 residues shorter linker lengths ($L_{onset}$). Assuming an extended conformation of the linker segment (~3 Å per residue), ADR1a folds ~9–12 Å deeper in the exit tunnel in uL23 Δloop ribosomes than in WT ribosomes.

## Spectrin and titin domains fold deeper in the exit tunnel in uL24 Δloop ribosomes

The 109-residue α-spectrin R16 domain has been shown to fold cotranslationally at $L \approx 35$ residues, in close proximity to uL24 in the exit port region (*Nilsson et al., 2017*). As seen in *Figure 3B* and *Table 1*, with both WT and uL23 Δloop ribosomes, R16 has $L_{onset}$ = 31 residues and $L_{max}$ = 35 residues (gray and red curves). For the uL24 Δloop ribosomes however, $L_{onset}$ = 29 residues and $L_{max}$ = 33 residues (*Table 1*), suggesting that that spectrin R16 folds ~6–7 Å deeper in the exit tunnel when the uL24 loop does not obstruct the tunnel exit port.

Similar results were obtained for the 89-residue titin I27 domain, *Figure 3C*. Previous studies have shown that the I27 domain folds at linker lengths $L$ = 35–39 residues and that it folds in about the same location as does spectrin R16, in close proximity to the uL24 loop (*Tian et al., 2018*). The $f_{FL}$ profile is not affected by the uL23 loop deletion, but folding commences at ~4 residues shorter linker lengths in uL24 Δloop ribosomes, similar to R16 (*Table 1*).

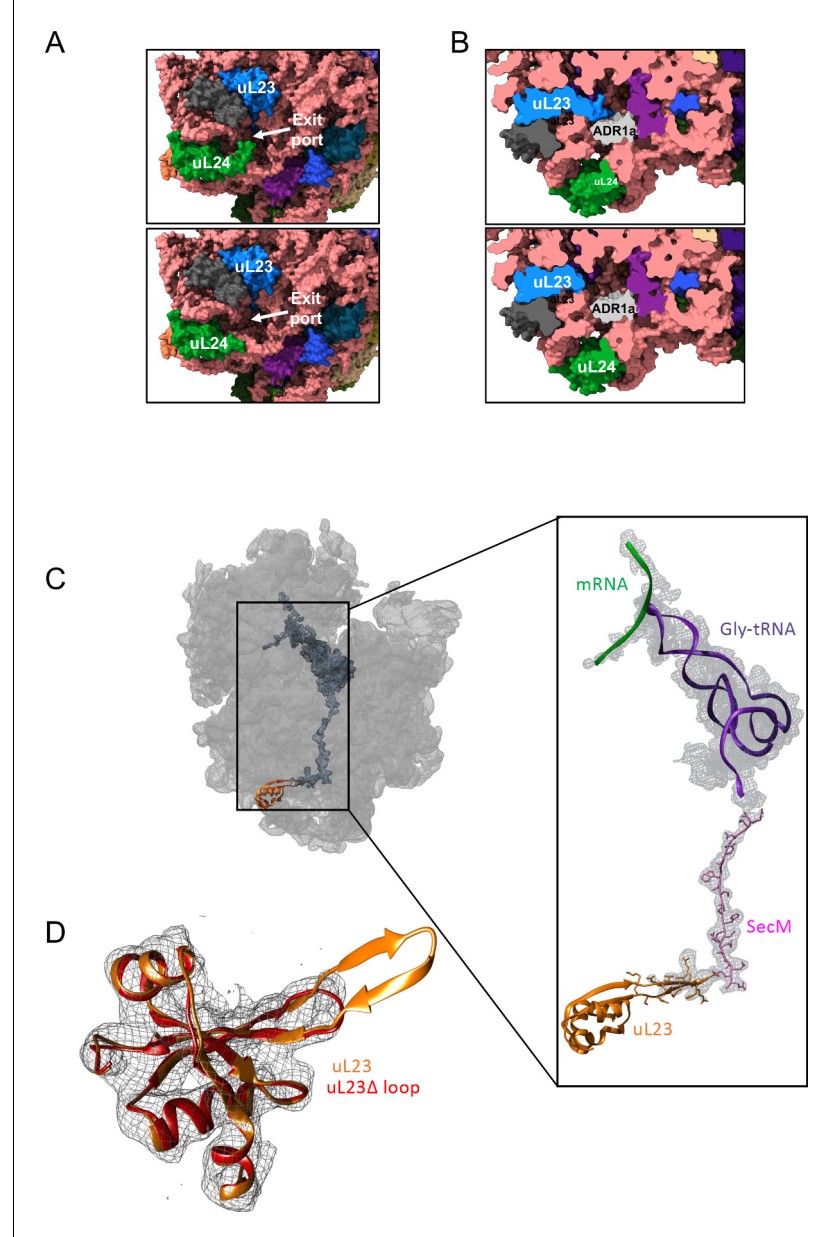

**Figure 2.** Structural consequences of removing the hairpin loops in uL24 and uL23 modeled after PDB 3JBU of the SecM stalled ribosome. (**A**) In wildtype ribosomes, the loop in uL24 partially obstructs the exit tunnel (top panel). Its removal in uL24 Δloop ribosomes creates a wide opening into the tunnel (bottom panel). (**B**) In wildtype ribosomes, the loop in uL23 extends into the exit tunnel (top panel). Its removal in uL23 Δloop ribosomes creates an open space around the area where the ADR1a domain is known to fold (*Nilsson et al., 2015*). The ADR1a structure is from PDB 5A7U. (**C**) Cryo-EM structure of the uL23 Δloop 70 S ribosome (EMD-4319), fitted to PDB 3JBU (that includes a Gly-tRNA and a 26-residue long arrested SecM AP) to locate uL23 (orange) and the exit tunnel. The enlarged region shows a difference map (in mesh) obtained by subtracting the cryo-EM map of the uL23 Δloop 70 S ribosome from a map generated from 3JBU in Chimera. The difference map shows that the only difference in volume between the two maps is the tRNA (in magenta), the SecM AP (in pink), and the loop deleted from uL23. (**D**) Extracted cryo-EM density (in mesh) for uL23 in the uL23 Δloop ribosome EMD-4319. Wildtype uL23 (orange) and a de novo-built model for the mutant uL23 Δloop protein (PDB 6FU8; red) are shown in ribbon representation.

DOI: https://doi.org/10.7554/eLife.36326.005

The following figure supplement is available for figure 2:

**Figure supplement 1.** Resolution map of the uL23Δ loop ribosome.

*Figure 2 continued on next page*

*Figure 2 continued*

DOI: https://doi.org/10.7554/eLife.36326.006

## Coarse-grained molecular dynamics simulations

In order to provide a more detailed structural framework for interpreting the $f_{FL}$ profile results, we performed coarse-grained molecular dynamics simulations of the cotranslational folding of ADR1a, spectrin R16, and titin I27 in WT, uL23 Δloop, and uL24 Δloop ribosomes, using a recently described model that allows us to calculate $f_{FL}$ profiles from the simulations (*Tian et al., 2018*). The essence of the method is that the simulations are used to determine folded and unfolded populations at each linker length, and the forces associated with them. Combining this information with the experimentally determined force-dependent escape rate of the AP from the ribosome (*Goldman et al., 2015*) in a kinetic model allows $f_{FL}$ to be calculated. Simulated (full lines) and experimental (dashed lines) $f_{FL}$ profiles are shown in *Figure 3D–F*, and detailed simulation results, together with representative snapshots from the simulations of the folded domains at $L \approx L_{onset}$, are shown in *Figure 3—figure supplement 10*. For all three proteins, the $L_{onset}$ values are well reproduced by the simulations, both in WT and Δloop ribosomes. $L_{max}$ values are also well captured by the simulations for ADR1a (WT ribosomes) and I27 (both WT and uL24 Δloop ribosomes), but are shifted to somewhat lower values in the R16 simulations.

The simulated ADR1a $f_{FL}$ profile for uL23 Δloop ribosomes, while showing an early onset of folding in agreement with the experimental profile, has a much smaller $L_{max}$ value. We also performed a simulation using a ribosome model with a smaller deletion in the uL23 loop (residues 70–72; red curve marked by X's); in this case, the peak in the simulated profile extends between $L_{onset}$ and $L_{end}$ values that are more similar to the experimental profile for uL23 Δloop ribosomes. The shape of the $f_{FL}$ profile for ADR1a is clearly highly sensitive to fine structural details of the exit tunnel and therefore somewhat difficult to reproduce by coarse-grained simulations.

In summary, both the experimental and simulation results are consistent with the idea that proteins start to fold as soon as they reach a part of the exit tunnel that is large enough to hold the folded protein. Judging from the $f_{FL}$ profiles, the 29-residue ADR1a domain folds approximately ~9–12 Å deeper in the exit tunnel in uL23 Δloop ribosomes than in WT and uL24 Δloop ribosomes, while the 89- and 109-residue titin and spectrin domains fold ~6–10 Å deeper inside the tunnel in uL24 Δloop ribosomes than in WT and uL23 Δloop ribosomes; the corresponding values estimated from the simulations are ~6 Å for ADR1a and ~13–15 Å for I27 and R16 (*Figure 3—figure supplement 10* panel B).

Both the uL23 and uL24 loops thus serve to reduce the space available for folding, but in different parts of the exit tunnel. The uL24 loop is particularly interesting in this regard. In bacterial ribosomes, it partially blocks the tunnel exit port, closing off what would otherwise be a wide, funnel-like opening, *Figure 2A*, and thereby prevents domains of $M_w \geq 10$ kDa from folding inside the exit tunnel. It is conserved (in length, if not in sequence) in bacterial ribosomes, *Figure 1—figure supplement 2*, suggesting that this will be the case not only for *E. coli* ribosomes but for bacterial ribosomes in general. Eukaryotic ribosome tunnels have different geometries owing to expansion segments in their rRNA as well as an increased number of proteins and a wider exit port (*Wilson and Doudna Cate, 2012*; *Filipovska and Rackham, 2013*); uL24 is among the most divergent proteins compared to bacteria (*Melnikov et al., 2015*). We therefore expect the precise relation between the onset of folding

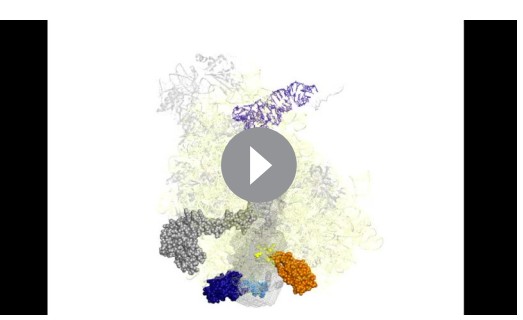

**Video 1.** The ribosome exit tunnel (mesh), as calculated for PDB 3JBU, uL23 Δloop and uL24 Δloop ribosomes by POVME. See *Figure 1A* for coloring scheme. The b-hairpin loops deleted in uL23 Δloop and uL24 Δloop ribosomes are shown in yellow and light blue, respectively. To facilitate the visualization of the exit tunnel, spheres left outside the exit tunnel after POVME processing were manually removed.

DOI: https://doi.org/10.7554/eLife.36326.007

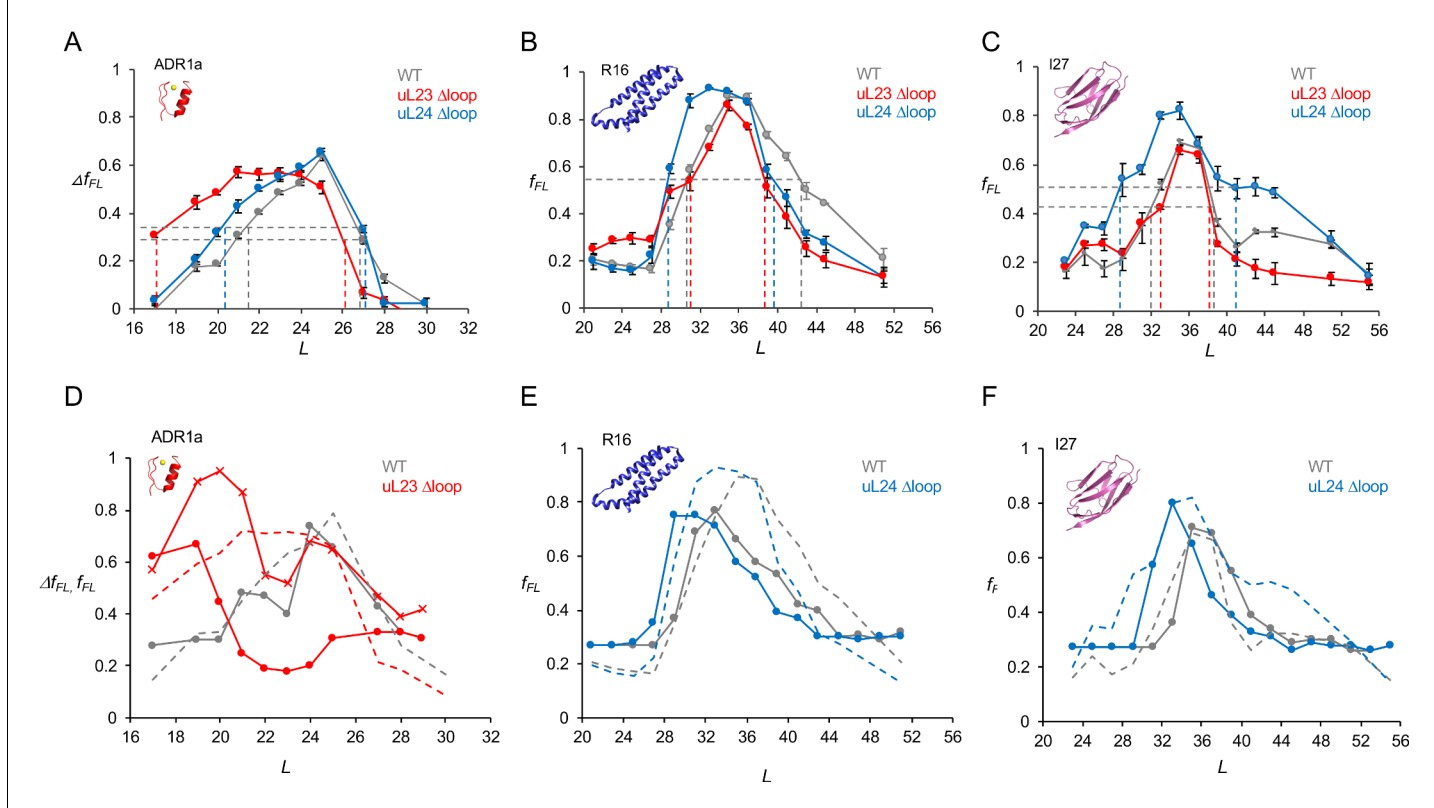

**Figure 3.** Cotranslational folding in WT, uL23 delta-loop, and uL24 delta-loop ribosomes. (A) $\Delta f_{FL}$ profiles ($\Delta f_{FL} = f_{FL}(50\ \mu M\ Zn^{2+}) - f_{FL}(50\ \mu M\ TPEN)$) for ADR1a constructs translated in the PURE system supplemented with in-house purified WT (gray), uL23 Δloop (red), and uL24 Δloop (blue) *E. coil* ribosomes. (B) $f_{FL}$ profiles for spectrin R16 constructs translated in the PURE system supplemented with in-house purified WT (gray), uL23 Δloop (red), and uL24 Δloop (blue) *E. coli* ribosomes. (C) $f_{FL}$ profiles for titin I27 constructs translated in the PURE system supplemented with in-house purified WT (gray), uL23 Δloop (red), and uL24 Δloop (blue) *E. coli* ribosomes. Error bars in panels a-c show SEM values calculated from at least three independent experiments. Dashed lines indicate $L_{onset}$ and $L_{end}$ values, c.f., **Table 1**. $f_{FL}$ profiles for non-folding mutants of R16 and I27 are found in (**Nilsson et al., 2017**; **Tian et al., 2018**). (D) Simulated $f_{FL}$ profiles (full lines) for ADR1a, spectrin R16, and titin I27 obtained with WT (gray), uL23 Δloop (red), and uL24 Δloop (blue) ribosomes. The corresponding experimental $f_{FL}$ profiles from panels a-c are shown as dashed lines. The simulated ADR1a $f_{FL}$ profile marked by X's was obtained with a uL23 Δloop(70-72) ribosome model. Simulated $f_{FL}$ profiles for ADR1a with uL24 Δloop ribosomes, and for R16 and I27 with uL23 Δloop ribosomes, are essentially identical to the corresponding profiles obtained with WT ribosomes, and are shown in **Figure 3—figure supplement 10**.

DOI: https://doi.org/10.7554/eLife.36326.008

The following source data and figure supplements are available for figure 3:

**Source data 1.** Experimental fFL values.
DOI: https://doi.org/10.7554/eLife.36326.019

**Figure supplement 1.** SDS PAGE showing ADR1 constructs translated in PURExpress Δ-Ribosome kit supplemented with high-salt-washed ribosomes isolated from HDB140, HDB143 (uL23 Δloop), or HDB144 (uL24 Δloop) as indicated.
DOI: https://doi.org/10.7554/eLife.36326.009

**Figure supplement 2.** SDS PAGE showing ADR1 constructs translated in PURExpress Δ-Ribosome kit supplemented with high-salt-washed ribosomes isolated from HDB140, HDB143 (uL23 Δloop), or HDB144 (uL24 Δloop) as indicated.
DOI: https://doi.org/10.7554/eLife.36326.010

**Figure supplement 3.** SDS PAGE showing ADR1 constructs translated in PURExpress Δ-Ribosome kit supplemented with high-salt-washed ribosomes isolated from HDB140, HDB143 (uL23 Δloop), or HDB144 (uL24 Δloop) as indicated.
DOI: https://doi.org/10.7554/eLife.36326.011

**Figure supplement 4.** SDS PAGE showing ADR1 constructs translated in PURExpress Δ-Ribosome kit supplemented with high-salt-washed ribosomes isolated from HDB140, HDB143 (uL23 Δloop), or HDB144 (uL24 Δloop) as indicated.
DOI: https://doi.org/10.7554/eLife.36326.012

**Figure supplement 5.** SDS PAGE showing Spectrin R16 constructs translated in PURExpress Δ-Ribosome kit supplemented with high-salt-washed ribosomes isolated from HDB140, HDB143 (uL23 Δloop), or HDB144 (uL24 Δloop) as indicated.
DOI: https://doi.org/10.7554/eLife.36326.013

*Figure 3 continued on next page*

*Figure 3 continued*

**Figure supplement 6.** SDS PAGE showing Spectrin R16 and Titin I27 constructs translated in PURExpress Δ-Ribosome kit supplemented with high-salt-washed ribosomes isolated from HDB140, HDB143 (uL23 Δloop), or HDB144 (uL24 Δloop) as indicated.

DOI: https://doi.org/10.7554/eLife.36326.014

**Figure supplement 7.** SDS PAGE showing titin I27 constructs translated in PURExpress Δ-Ribosome kit supplemented with high-salt-washed ribosomes isolated from HDB140, HDB143 (uL23 Δloop), or HDB144 (uL24 Δloop) as indicated

DOI: https://doi.org/10.7554/eLife.36326.015

**Figure supplement 8.** $f_{FL}$ profiles for ADR1a constructs translated in PURE by WT, uL23 Δloop, and uL24 Δloop ribosomes, either in the presence of $Zn^{2+}$ or of the $Zn^{2+}$ chelator TPEN.

DOI: https://doi.org/10.7554/eLife.36326.016

**Figure supplement 9.** Sequences of the longest and the shortest constructs used for each protein and, a depiction of the location of the sequences the ribosome exit tunnel when the last residue of the AP is in the P-site (lower panel, yellow box).

DOI: https://doi.org/10.7554/eLife.36326.017

**Figure supplement 10.** Summary of results from coarse-grained MD simulations.

DOI: https://doi.org/10.7554/eLife.36326.018

and protein $M_w$ to be somewhat different in eukaryotic ribosomes, as also suggested by a recent study (*Schiller et al., 2017*). At present, we do not know to what extent the shape of the ribosome exit tunnel has evolved to optimize the conditions for cotranslational protein folding in different organisms and organelles, but it is not unlikely that such a connection exists.

# Materials and methods

## Key resources table

| Reagent type (species) or resource | Designation | Source or reference | Identifiers | Additional information |
|---|---|---|---|---|
| Strain, strain background (*Escherichia coli*) | HDB140, HDB143, HDB144, Strain background N281 | 10.1111/j.1365–2958.2010.07325.x | NA | Strains used to isolate high-salt washed ribosomes in this study. |
| Antibody | uL23, uL24 | 10.1111/j.1365–2958.2010.07325.x | NA | 1:8000 dilution used (incubated for one hour). Secondary antibody: Mouse (1:20,000 dilution incubated for one hour). Nitrocellulose membrane blocked with 5% Milk in TBS-T for an hour. |
| Peptide, recombinant protein | uL23 Δ loop, uL24 Δ loop | 10.1111/j.1365–2958.2010.07325.x | UniProtKB- P0ADZ0 (rplW) UniProtKB-P60624 (rplX) | Referred to as HDB 143 and HDB 144 in original paper. Refers to genes rplWΔ65–74 and rplX Δ43–57 respectively. |
| Commercial assay or kit | GeneJET Plasmid miniprep kit | Thermo Fisher Scientific RRID:SCR_008452 | Cat no. K0502 | Used to purify plasmids |
| Commercial assay or kit | PURExpress Δ Ribosome kit | New England Biolabs | Cat no. E3313S | Kit was supplemented with ribosomes purified in the lab. Translation carried out at 37°C for 20 min. |
| Software, algorithm | EasyQuant | doi: 10.1038/nsmb.2376 | NA | Used to quantify relative fraction full-length of translated protein from SDS-PAGE |

*Continued on next page*

*Continued*

| Reagent type (species) or resource | Designation | Source or reference | Identifiers | Additional information |
|---|---|---|---|---|
| Software, algorithm | cryoSPARC version v2 | Structura Biotechnology Inc | NA | Used for ab-initio reconstruction of uL23Δloop ribosomes. The following operations were carried out as part of the cryoSPARC workflow: 2D classification, Ab initio reconstruction, Homogeneous refinement, Sharpening and map flipping to correct for handedness, local resolution. |
| Software, algorithm | UCSF ChimeraX | | SCR_015872 | Used to make *Figure 2A* |
| Software, algorithm | UCSF Chimera v. 1.12 | J Comput Chem. 2004 Oct;25 (*Nilsson et al., 2015*): 1605–12. | SCR_004097 | Used to visualise the cryoEM map, fit PDB models 3JBU, 4YBB to check for differences in maps. Used to make figures. |
| Software, algorithm | Jalview v 2.10.4 | doi: 10.1093/ bioinformatics/btp033 | SCR_006459 | Use for generating multiple sequence alignments of uL23 and uL24 in the supplementary figures. |
| Chemical | Potassium acetate | Sigma-Aldrich (SCR_008988) | Cat no. P1190 | Source of potassium ions to stabilise ribosomes |
| Chemical | Magnesium acetate | Sigma-Aldrich (SCR_008988) | Cat no. M5661 | Source of Magnesium ions to stabilise ribosomes |
| Chemical | cOmplete protease inhibitor cocktail | Sigma-Aldrich (SCR_008988) | Cat no. 04693116001 | Used as a protease inhibitor during cell lysis to obtain ribosomes |
| Chemical | N,N,N′,N′-Tetrakis (2-pyridylmethyl) ethylenediamine | Sigma Aldrich (SCR_008988) | Cat no. P4413 | Used to chelate Zn for the -Zn reactions in the ADR1 plot. |
| Chemical | threo-1,4-Dimercapto-2, 3-butanediol DL-Dithiothreitol | Sigma Aldrich (SCR_008988) | Cat no. DTT-RO | Reducing agent added to buffers during ribosome purification and as a reductant for SDS-PAGE |
| Chemical | Tris Base | Sigma-Aldrich (SCR_008988) | Cat no. T1503 | Buffering agent during ribosome preparation |
| Chemical | 35S Methionine | Perkin-Elmer | Cat no. NEG009T001MC | 35S Methionine is incorporated into the protein during in vitro translation and aids detection by phosphorimaging. |

## Enzymes and chemicals

The PURExpress Δ Ribosome kit was purchased from New England Biolabs (Cat no. E3313S). The components used to prepare Lysogeny Broth (LB Medium) for ribosome isolation were obtained

**Table 1.** $L_{onset}$, $L_{max}$, and $L_{end}$ values calculated from the $f_{FL}$ profiles in **Figure 3**.

| | ADR1a | | | R16 | | | I27 | | |
|---|---|---|---|---|---|---|---|---|---|
| | WT | uL23 Δloop | uL24 Δloop | WT | uL23 Δloop | uL24 Δloop | WT | uL23 Δloop | uL24 Δloop |
| $L_{onset}$ | 21 | 17 | 20 | 31 | 31 | 29 | 32 | 33 | 28 |
| $L_{max}$ | 25 | 22 | 25 | 35 | 35 | 33 | 35 | 35 | 35 |
| $L_{end}$ | 27 | 26 | 27 | 42 | 39 | 40 | 38 | 38 | 41 |

DOI: https://doi.org/10.7554/eLife.36326.020

from BD Biosciences and all other chemicals used were sourced from Merck Sigma Aldrich. ($^{35}$S) Methionine was purchased from Perkin Elmer. Bis-Tris gels and plasmid isolation kits were obtained from Thermo Scientific.

## Plasmids

All ADR1, spectrin and titin constructs fused to the *E. coli* SecM AP via a variable linker were expressed from the pET19b vector, as described previously (*Nilsson et al., 2015*; *Nilsson et al., 2017*; *Tian et al., 2018*). The spectrin constructs used in this study lacked the soluble domain of LepB at the N-terminus.

## Strains and antisera

Strains HDB140 (referred to as WT), HDB143 (referred to as uL23 Δloop) and HDB144 (referred to as uL24 Δloop), as well as rabbit polyclonal antisera against uL23 and uL24, are described in (*Peterson et al., 2010*).

## Isolation of ribosomes

Ribosomes were purified from the strains HDB140, HDB143, and HDB144. The strains were cultured in Lysogeny broth (LB) to an $A_{600}$ of 1.0 at 37°C and chilled on ice for 15 min before they were harvested by centrifugation at 4000 g for 10 min. The cell pellet was washed twice with Buffer A at pH 7.5 (10 mM Tris-OAc, 14 mM Mg(OAc)$_2$, 60 mM KOAc, 1 mM DTT, 0.1% Complete Protease Inhibitor) and lysed using the Emulsifex (Avestin) at a pressure of 8000 psi. The cell lysate was loaded on a sucrose cushion at pH 7.5 (50 mM Tris-OAc, 1 M KOAc, 15 mM Mg(OAc)$_2$, 1.44 M sucrose, 1 mM DTT, 0.1% Complete Protease Inhibitor) and centrifuged at 80,000xg in a Ti70 rotor (Beckman-Coulter) for 17 hr. The obtained ribosomal pellet was resuspended in Buffer B at pH 7.5 (50 mM Tris-OAc, 50 mM KOAc, 5 mM Mg(OAc)$_2$, 1 mM DTT), flask frozen in liquid nitrogen and stored at −80°C. This suspension of ribosomes is presumed to consist of a pool of non-translating 30S, 50S and 70S particles due to the concentration of Mg$^{2+}$ in the buffer they are in. Each batch of ribosomes that was prepared was tested for optimal translation by titrating different volumes in the PURExpress Δ-Ribosome kit.

## In vitro transcription and translation

The generated constructs were translated for 20 min. in the PURExpress Δ-Ribosome kit supplemented with high-salt-washed ribosomes isolated from HDB140, HDB143 (uL23 Δloop), or HDB144 (uL24 Δloop). Plasmid DNA of each construct (300 ng) was used as a template for polypeptide synthesis, and translation was carried out in the presence of ($^{35}$S) Methionine at 37°C for 20 min and shaking at 500 r.p.m. For ADR1a constructs, the translation reactions also included either 50 µM zinc acetate or 50 µM of the Zn$^{2+}$ chelator TPEN. Translation was stopped by treating the sample with a final concentration of 5% trichloroacetic acid (TCA) and incubated on ice for 30 min. The TCA precipitated samples were subsequently centrifuged at 20,000 g for 10 min in a tabletop centrifuge (Eppendorf) and the pellet obtained was solubilized in sample buffer, supplemented with RNaseA (400 µg/ml), and incubated at 37°C for 15 min. The samples were resolved on 12% Bis-Tris gels (Thermo Scientific) in MOPS buffer for ADR1 and MES buffer for Spectrin and Titin. Gels were dried and subjected to autoradiography and scanned using the Fujifilm FLA-9000 phosphorimager for visualization of radioactively labeled translated proteins.

## Quantification of radioactively labelled proteins

The protein bands on the gel were quantified using MultiGauge (Fujifilm) from which one-dimensional intensity profiles of each gel lane was extracted. This information was subsequently fit to a Gaussian distribution using EasyQuant (Rickard Hedman, Stockholm University). The sum of the arrested and full-length bands was calculated, and this was used to estimate the fraction full-length protein for each construct.

## Cryo-EM sample preparation and data processing

The uL23 Δloop ribosomes (4 $A_{260}$/ml) diluted in grid buffer (20 mM HEPES-KOH, 50 mM KOAc, 10 mM Mg(OAc)$_2$, 125 mM sucrose, 2 mM Trp, 0.03% DDM) were loaded on Pelco TEM 400 mesh Cu

grids pre-coated with 2 nm thick carbon and frozen using the Vitrobot Mark IV (FEI). Data were collected on the Titan Krios (FEI) microscope operated at 300 keV and equipped with a Falcon II direct electron detector. The camera was set to a nominal magnification of 75,000X, which resulted in a pixel size of 1.09 Å at the sample level and a defocus range of −1 to −3 μm.

The frame dose used was 1.17 e/Å$^2$, and 20 frames were aligned using MotionCor2 (*Li et al., 2013*) within the Scipion software suite (*de la Rosa-Trevín et al., 2016*). The micrographs were visually inspected and those within a resolution threshold of 5 Å were selected, yielding 3522 micrographs.

471,272 particles were picked using Xmipp manual-pick followed by particle extraction within Scipion and further processing in CryoSPARC (*Punjani et al., 2017*). Two rounds of 2D classification were done, and particles resembling 30S and 50S subunits alone were discarded after visual inspection of the classes. The remaining 297,363 particles of the 70S ribosome were subjected to *ab initio* reconstruction into three classes to further sort out heterogeneity. A single homogeneous class consisting of 132,029 particles was used for final homogeneous refinement that resulted in a final map with an average FSC resolution at 0.143 of 3.28 Å. The obtained map of the 70S ribosome was sharpened and corrected for handedness in CryoSPARC fitted with PDBs 3JBU and 4YBB in Chimera (*Pettersen et al., 2004*). Local resolution and FSC at 0.143 was estimated in cryoSPARC. The electron microscopy map was deposited in the Electron Microscopy Data Bank.

The initial model for uL23 Δloop was built with Coot, and improved by energy minimization in a solvated dodecahedron box of explicit TIP3P waters, neutralized with chloride ions and using the Amber 99SB-ILDN force field (*Lindorff-Larsen et al., 2010*). The steepest descent minimization method implemented in GROMACS 2016.1 was used (*Abraham et al., 2015*; *Pal et al., 2014*). Even after minimization, the backbone of the new loop formed after the deletion of residues 65–74 still showed improper geometry and Ramachandran outliers, so we used kinematic sampling (*Bhardwaj et al., 2016*) to model alternative loop conformations, and then we selected the loop that could fit the electron density and had the best Ramachandran score.

Figures were prepared using MacPymol 1.8.6.2 (Schrödinger LLC), Chimera (*Pettersen et al., 2004*), and ChimeraX (*Goddard et al., 2018*).

## Calculation of tunnel volume

The volume calculations were performed with POVME 2.0 (1). We used the *E. coli* SecM structure PDB 3JBU as a reference. To determine the inclusion region, we generated a series of overlapping spheres- eight with a 20 Å radius, and one with a 40 Å radius. In order to have a complete coverage of the exit tunnel, the centers of the spheres were chosen to match the coordinates corresponding to alternating Cα atoms of the amino acids of the SecM arrest peptide located within the exit tunnel (for the 20 Å radius spheres the residues use as centers were D11, F13, T15, V17, I19, Q21, Q23, I25, A27, G28 and for the 40 Å radius sphere the residue was E3). Grid Spacing was set to 2.0 Å, and the distance cut-off to 1.09 Å. For all three cases (WT, uL24 Δ loop, uL23 Δ loop), we used the same inclusion region. We also removed the SecM arrest peptide located within the exit tunnel. For uL23 Δloop ribosomes residues 65–75 were removed from uL23, and for uL24 Δloop ribosomes residues 42–57 were removed from uL24 (numbering based on PDB 3JBU) prior to the calculation.

## Kinetic model to calculate fraction full length protein f$_{FL}$(t)

The theoretical force profiles (*Figure 3D–F*) for ADR1a, I27, and R16 were calculated based on a kinetic model introduced in our previous study (*Tian et al., 2018*). Briefly, the rate, $k_e$, of the arrest peptide sequence escape from the peptidyl transfer center with a force ($F$) exerted by the folding protein can be calculated using the Bell model:

$$k_e(F) = k_0 e^{F\Delta x^{\ddagger}/k_B T},$$

where $\Delta x^{\ddagger}$ is the distance from the free energy minimum to the transition state, $k_0$ the rupture rate when force equals to zero, $k_B$ is Boltzmann's constant, and $T$ the absolute temperature. In this study, $k_0$ and $\Delta x^{\ddagger}$ are set to be $3.4 \times 10^{-4}$ s$^{-1}$ and 4.5 Å, respectively, based on a previous experimental study (*Goldman et al., 2015*) in which $k_0$ and $\Delta x^{\ddagger}$ were estimated to be in the range of $0.5 \times 10^{-4}$ to $20 \times 10^{-4}$ s$^{-1}$ and 1-8 Å, respectively.

We assume that the folding and unfolding of the protein is much faster than the escape from the ribosome. Then the time-dependent force profile $f_{FL}(t)$ can be obtained approximately by the mean pulling forces exerted when the protein is unfolded, $F_u$, or folded, $F_f$, and the unfolded and folded populations of $P_u$ and $P_f$ respectively,

$$f_{FL}(t) \approx 1 - \exp[-t[P_u k_e(F_u) + P_f k_e(F_f)]].$$

Note that the values of $F_u$, $F_f$, $P_u$, $P_f$ are dependent on the linker length *L*, and can be determined by molecular dynamics simulations.

## Molecular dynamics simulations of ribosome-nascent chain complex

A coarse-grained model was employed to simulate folding of ADR1a, titin I27, and spectrin R16 on the ribosome. The ribosome was modelled on the 50S subunit of the *E. coli* ribosome (PDB 3OFR; *Dunkle et al., 2010*) Each amino acid in the nascent chain and ribosome was represented by one bead at the position of the α-carbon atom, each RNA residue was modelled by three beads located at the positions of phosphate P, sugar C4', and base N3 atoms (*Voss and Gerstein, 2005*). The uL23 Δloop ribosome was modelled by replacing the coordinates of the wildtype uL23 protein with the cryo-EM structure of the uL23 Δloop protein from this study (PDB 6FU8), after being aligned to the wild type protein. The uL24 Δloop ribosome was modelled by replacing the coordinates of the wildtype uL24 protein with the structure of the uL24 Δloop protein built by homology modelling with Modeller (*Eswar et al., 2006*).

The interactions within the nascent chain were governed by a standard structure-based model (*Karanicolas and Brooks, 2002*), which allowed it to reversibly fold to the native state and unfold. Interactions between the protein and ribosome beads were purely repulsive. The pulling force (F) exerted on the arrest peptide by the folding of the protein (ADR1a, R16, or I27) was measured by the extension of the harmonic pseudobond potential between the last and the second last amino acid of the SecM arrest peptide. More details can be found in our previous study (*Tian et al., 2018*).

## Additional information

### Funding

| Funder | Grant reference number | Author |
| --- | --- | --- |
| Knut och Alice Wallenbergs Stiftelse | 2012.0282 | Gunnar von Heijne |
| Vetenskapsrådet | 621-2014-3713 | Gunnar von Heijne |
| Cancerfonden | 15 0888 | Gunnar von Heijne |
| National Institutes of Health | Intramural | Robert B. Best Harris D Bernstein |
| National Institutes of Health | R35GM122543 | Fátima Pardo-Avila |

The funders had no role in study design, data collection and interpretation, or the decision to submit the work for publication.

### Author contributions

Renuka Kudva, Conceptualization, Formal analysis, Investigation, Methodology, Writing—original draft, Writing—review and editing; Pengfei Tian, Conceptualization, Investigation, Visualization, Methodology, Writing—original draft, Writing—review and editing; Fátima Pardo-Avila, Investigation, Methodology, Writing—original draft; Marta Carroni, Investigation, Methodology; Robert B Best, Conceptualization, Formal analysis, Supervision, Funding acquisition, Writing—original draft, Writing—review and editing; Harris D Bernstein, Conceptualization, Formal analysis, Funding acquisition, Writing—original draft, Writing—review and editing; Gunnar von Heijne, Conceptualization, Formal analysis, Funding acquisition, Investigation, Methodology, Writing—original draft, Project administration, Writing—review and editing

## Author ORCIDs

Renuka Kudva (iD) http://orcid.org/0000-0003-0426-3716
Marta Carroni (iD) http://orcid.org/0000-0002-7697-6427
Robert B Best (iD) https://orcid.org/0000-0002-7893-3543
Harris D Bernstein (iD) http://orcid.org/0000-0002-4941-3741
Gunnar von Heijne (iD) http://orcid.org/0000-0002-4490-8569

## Decision letter and Author response

Decision letter https://doi.org/10.7554/eLife.36326.027
Author response https://doi.org/10.7554/eLife.36326.028

## Additional files

### Supplementary files

• Transparent reporting form
DOI: https://doi.org/10.7554/eLife.36326.021

### Data availability

The Cryo-EM map has been deposited under accession code EMD-4319, and the Atomic model for uL23 Dloop deposited under PDB accession number 6FU8.

The following datasets were generated:

| Author(s) | Year | Dataset title | Dataset URL | Database and Identifier |
|---|---|---|---|---|
| Kudva R, von Heijne G | 2018 | E.coli ribosome with Δ-hairpin of uL23 deleted | https://www.ebi.ac.uk/pdbe/entry/emdb/EMD-4319 | EMDataBank, 4319 |
| Kudva R, von Heijne G | 2018 | Atomic model for uL23 delta-loop | http://www.rcsb.org/structure/6FU8 | Protein Data Bank, 6FU8 |

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
