## [Decision Letter]

[Editors’ note: this article was originally rejected after discussions between the reviewers, but the authors were invited to resubmit after an appeal against the decision.]

Thank you for submitting your work entitled "The Shape of the Ribosome Exit Tunnel Affects Cotranslational Protein Folding" for consideration by *eLife*. Your article has been reviewed by three peer reviewers, and the evaluation has been overseen by John Kuriyan as the Senior and Reviewing Editor. The reviewers have opted to remain anonymous.

Based on discussions with the reviewers and the individual reviews below, we regret to inform you that your work will not be considered further for publication in *eLife*. The reviewers and the editors agree that this paper is an interesting and important study of the folding of nascent peptides, and how the shape of the ribosome exit tunnel governs that folding. Despite the interest in the work, after discussion with the reviewers, we have concluded that the paper is not quite definitive enough in its conclusions for further consideration at *eLife*. We remain interested in the work, and if you are able to provide additional data to support the conclusions and to address the reviewer comments satisfactorily, we would welcome a resubmission. We recognize, however, that you might wish to submit the manuscript elsewhere at this point.

The three individual reviews are appended below.

*Reviewer 1:*

This paper demonstrates the influence that changes in geometry of the peptide exit tunnel can have on the timing of co-translational protein folding. The authors conduct in vitro translation assays to translate folding domain fused to the SecM stalling sequence with varying linker length to measure at which linker lengths the nascent chain starts to fold on the ribosome, applying a tugging force to rescue the stall and translate the full protein. After using Cryo-EM to confirm that the loop deletion mutations on uL23 and uL24 doesn't change the architecture of the peptide exit tunnel, the authors indeed show that the loop deletions alter where the nascent chain folding occurs in the tunnel. Overall, the findings show a straightforward method to probe the location within the peptide exit tunnel where the protein folding occurs, and is indeed an advance in the field. The novelty here is in the manipulation of the tunnel itself, and the correlation to structural data. However, the question remains whether the science presented here rises to the level of publication in *eLife*. There are some critiques that the authors should address outlined below:

1) It is not explicitly mentioned whether the uL23Δloop ribosomes prepared for Cryo-EM is a 50S, empty 70S, or stalled translation complexes. This needs to be made in the Materials and methods section under the "Cryo-EM sample preparation and data processing" part.

2) Why is there an "increase in fFL seen at L = 17 residues in the presence of TPEN"? What does TPEN do to cause the increase in fFL? A brief statement in the text commenting on this effect would be helpful. If indeed this is an effect of TPEN, is there a way to do the same experiment in the absence of Zn^2+^ or TPEN to confirm this?

3) The authors commented on the universality of the uL24 loop in bacteria. It would be great to also comment on the non-universality of uL23 or uL24 loops in other domains of life (like the absence of uL23 loop in spinach chloroplast ribosomes mentioned in the paper) and how it may suggest that co-translational protein folding may work differently in another organism due to differences in geometry of its ribosomal peptide exit tunnel.

*Reviewer 2:*

Kudva et al. investigate how the geometry of the ribosomal tunnel affects folding. The main intent of this study is to understand how modifications on the ribosome influence the location of co-translational folding of proteins of different sizes. The authors first describe the folding assay; briefly, the force that protein folding close to the exit vestibule can exert on the SecM stalling sequence can consequently cause ribosome readthrough the mRNA and translate additional residues.

The authors explain the basis of the modified ribosomes HDB143 (ΔuL23) and HDB144 (ΔuL24) and solve the cryoEM structure of ΔL23 ribosomes. They further characterise the structural differences between wt and mutant by comparing the tunnel volumes. The authors then analyse the force profile of the ADR1, spectrin and titin domains on the wt, ΔL23 and ΔL24 ribosomes in an attempt to dissect the contribution of the two loops from these proteins in protein folding. They observe that ADR1 folds in the exit tunnel in the absence of loop in uL23 but not uL24, whereas the titin and spectrin (R15, R16) domains fold inside the tunnel in ΔL24 rather than the wt and ΔL23.

This study attempts to deconvolute the tunnel components that may be important for co-translational folding using the force assay which the authors are known for. This is a valuable, complex and rich experiment. It seems to me, however, not enough to interpret such complex events of translational and folding. This is my main concern: all conclusions are drawn based on the force experiments alone. Also, although some structural data are presented they are not used to rationalize the force results. It seems therefore the authors try to overstretch interpretation of their data.

Overall the manuscript contains some nice experiments and this is an interesting study. The study however, seems to need more substance to support some of its more far reaching statements. For example, the broader implications at the end would be excellent, had the paper have more depth. For me, this study in its current form leaves too many open questions.

1) Can the authors comment on the experimental conditions used for translation. Do the mutant ribosomes have the same translational kinetics as for wt? Are any differences likely to influence the force plots?

2) The cryoEM structure of the uL23 ribosomes is interesting, but why the authors did not investigate programmed ribosomes as they have previously? However, using the uL23 structure, can the authors comment further on the structural environment that the ADR1a would have in the uL23 ribosome compared to WT ribosomes? Does it have more space than in the original location? Given the rationale that the authors use to describe where folding begins to takes place (e.g. ca. 17aa in uL23 deletion). More information regarding the significance of structure would help.

3) It is not very clear to me how the authors define folding. Descriptions of the plots include words like "starting to increase, starting to fold". More information is needed on how these points are determined because to me it seems very subjective. In the absence of unfolded controls (e.g. in R15 and R16), the context for the "dips" and "rise/peaks" are difficult to rationalise.

4) On a similar point, overall, the analysis of Figure 3 appears ambiguous. For example, in Figure 3B red (uL23), data points from L=21-24 are effectively identical, all in error range, which casts potential doubt on the contention that there is a shift by 2-3 residues. The same types of difficulties arise in following analysis Figure 3D and 3E. One cannot clearly say there is 2-3 residue shifts. In order to focus on this small difference between the profiles, there should be more detailed analysis on other differences between the force profiles, for instance, why there are significant differences in fFL values between different ribosomes with the same length of nascent chains.

5) In ADR1a in the ΔuL23 ribosomes the first point (17aa) is non-zero even after subtracting the unfolded control. This suggests that the ADR1a is beginning to fold 51Ang (3ang per aa) from the PTC, approximately near uL22. If this is a folding, why is there never a clear peak? The authors say that the EM data tells us that the tunnel hasn't changed – even if the uL23 loop is gone and folding 'begins' early, (a) why isn't a complete fold reached (b) why would the ADR1a complete its fold in a narrow region of the tunnel rather than the vestibule exit? The "spring constant" idea is attractive – if the tunnel has opened up, then the NC can begin to sample conformations immediately, and in this case never reaches a maximum point of tension in the tunnel.

6) There are differences in Figure 1B observed between wt ribosomes in the PURE-kit or those prepared n-house and no explanation is given as to why? The WT and the WT(purified) ribosomes also show different patterns, even if the peak appears to be retained. Can the authors comment on the general reproducibility? Additionally, the PURE-derived force profile has no error bars – were replicates performed? Why is Δf used solely for this graph?

7) The use of unfolded controls for ADR1a shows clearly need to subtract data – why was this not done for R15, R16 and I27 plots?

8) Overall, the inclusion of both R15 and R16 is perhaps somewhat strange: on one hand, analysis of R15 is more convincing that other nascent chains are used to elaborate on the differences in force profiles. However, can the authors elaborate further on the significance of R15 and R16 and how these results relate to their previous work? For the R15 force profile, the peak of the wild type does not look the same as the previously reported, as it has a peak of 37 amino acids. Although the authors explain that this is because a folding intermediate is formed this is not clear or supported by other data. They also attribute the observed difference in force amplitude between wt and ΔuL23 to the lower "spring constant" due to the wider tunnel in the latter. The discussion of a 'spring tension' was interesting, but not explored and it seemed a pity in this context. Again, supporting data are needed.

9) The folding intermediate proposed in R15 – why would this intermediate only exist in uL23 deleted and WT ribosomes, but not in uL24 deleted ribosome? This seemed to be unclear. If the protein off the ribosome is renatured from an unfolded state does it form intermediates on its folding pathway?

10) It is not clear where in the ribosome the 6-9 Å measurement corresponds to in both the ΔuL23 ribosomes (subsection “ADR1a folds deeper in the exit tunnel in uL23 Δloop but not in uL24 Δloop ribosomes”, last paragraph) and in ΔuL24 ribosomes (subsection “Spectrin and titin domains fold deeper in the exit tunnel in uL24 Δloop ribosomes”, third paragraph). How does this correlate to the structure – i.e. is it consistent with the approximate location for the uL23 deletion?

11) The concluding statements appear to extrapolate from the data too much, and don't appear to be supported by the experimental data presented in the study – for example: "may prevent premature folding".…."that depend on interactions with TF…" Similarly, the uL24 loop statements about correlating length and sequence is not particularly well supported either – it seems to be a rather loose conclusion to an otherwise interesting paper.

*Reviewer 3:*

This manuscript reports a study of cotranslational folding, with the major conclusion that the shape of the exit tunnel modulates the cotranslational folding process. This conclusion is reached by studying two different proteins and two different forms of the ribosome, with the clever idea of truncating parts of L23 and L24.

My major concern with this manuscript is that the data presented do not adequately support the conclusions. The only experimental technique used is an assay in which of a SecM translational arrest peptide acts, broadly speaking, as a force sensor, whereby the measurements are performed by SDS-PAGE.

It is very difficult to assess the errors associated with this assay, in part because the primary data are not reported.

Most importantly, however, these conclusions should be validated using an independent technique. In the absence of this validation, one should put an excessive trust in the assay used, which would seem to be low resolution at best.

Overall, the publication of this work would be premature.

[Editors’ note: what now follows is the decision letter after the authors submitted for further consideration.]

Thank you for submitting your article "The Shape of the Ribosome Exit Tunnel Affects Cotranslational Protein Folding" for consideration by *eLife*. Your article has been reviewed by two peer reviewers, and the evaluation has been overseen by John Kuriyan as the Reviewing Editor and Senior Editor. The reviewers have opted to remain anonymous.

Your paper is now, in principle, acceptable for publication, but we are asking for some minor revisions (see below).

Your paper was seen by two of the original reviewers. One of the reviewers approves the revised manuscript. The other reviewer continues to object to the experimental basis for the paper, stating:

"The issue here is whether a single technique can be used to draw conclusions in a high impact paper. I am afraid that I believe that the answer is negative. I take the point made by the authors that this technique has been validated on other systems, but the fact remains that it is not as high resolution and clear cut as other techniques, such as cryo-EM or NMR.

Although I appreciate the quality and the scope of this manuscript, I do not think that the reliability of the conclusions meets the standard required by *eLife*."

We are making an editorial decision to proceed with publication of your paper at *eLife*. As someone outside the field, I find the entire body of work describing co-translational folding (either inside the ribosome or in its vicinity) to be extremely interesting, and yielding results that could not have been anticipated a few years ago. Although the results may now seem obvious to people in the field, I feel that the results are still very intriguing. I also note that there are controversies, but this reflects the complexity of the phenomena, and the limited view provided by different experimental techniques. Time will sort this out.

Regarding the objection that the paper relies on only one experimental technique, we accept your argument that this method (protease sensitivity with arrest peptide fusions) has been sufficiently validated in past work. While it would be nice to have the results of other experimental probes, we recognize that you have submitted this work as a short report. Also, I am not sure that NMR and cryoEM necessarily provide definitive answers in cases where a dynamic process is being observed. They too are subject to artifacts arising from restrictions on what can be observed.

Before publication, we encourage you to include in the paper more context for the importance and significance of the work, such as provided in your recent PNAS paper (Farias-Rico et al., 2018). We also suggest that you include some additional explanation for the validation of the method, as you suggest in your response to reviewers. You may do this as you think best suits your purposes, and we will approve your revised manuscript for publication without further delay.

---

## [Author Response]

[Editors’ note: the author responses to the first round of peer review follow.]

Thanks for the thorough review of our manuscript. We take heart in the fact that you say that you’re still interested in the work, despite the rejection (which we understand, given some of the reviewer comments).

Our reading of the reviewer reports (especially reviewers 2 and 3) suggests that the main critique of the manuscript in its present form is that we use only the arrest-peptide based force-measurement assay in the paper, an assay that reviewer 3 in particular feels is not well validated against other techniques for studying cotranslational folding. As detailed in the response to reviewer 3 below, we (emphatically…) do not agree that the technique has not been properly validated, but realize that we should have been more explicit in summarizing previously published data to address this issue.

The second general issue we want to raise is that we wrote this paper in an attempt to make a single, major point based on just one experiment (well, one experiment for each protein domain and each type of mutant ribosome) and therefore, following the *eLife* Author Instructions, submitted it as a Short Report. Judging from their comments, we’re not sure that the reviewers appreciated that it wasn’t intended as a full paper.

Reviewer 1:

This paper demonstrates the influence that changes in geometry of the peptide exit tunnel can have on the timing of co-translational protein folding. The authors conduct in vitro translation assays to translate folding domain fused to the SecM stalling sequence with varying linker length to measure at which linker lengths the nascent chain starts to fold on the ribosome, applying a tugging force to rescue the stall and translate the full protein. After using Cryo-EM to confirm that the loop deletion mutations on uL23 and uL24 doesn't change the architecture of the peptide exit tunnel, the authors indeed show that the loop deletions alter where the nascent chain folding occurs in the tunnel. Overall, the findings show a straightforward method to probe the location within the peptide exit tunnel where the protein folding occurs, and is indeed an advance in the field. The novelty here is in the manipulation of the tunnel itself, and the correlation to structural data. However, the question remains whether the science presented here rises to the level of publication in eLife. There are some critiques that the authors should address outlined below:1) It is not explicitly mentioned whether the uL23Δloop ribosomes prepared for Cryo-EM is a 50S, empty 70S, or stalled translation complexes. This needs to be made in the Materials and methods section under the "Cryo-EM sample preparation and data processing" part.

The cryo-EM was done with non-translating 70S ribosomes. This is now stated and highlighted in the Materials and methods section.

2) Why is there an "increase in fFL seen at L = 17 residues in the presence of TPEN"? What does TPEN do to cause the increase in fFL? A brief statement in the text commenting on this effect would be helpful. If indeed this is an effect of TPEN, is there a way to do the same experiment in the absence of Zn^2+^ or TPEN to confirm this?

In the *L*=17 construct, ADR1a is appended directly upstream of the SecM arrest peptide (Figure 3—figure supplement 9). Although we haven’t tested this, we believe that the slight increase in *f_FL_* seen in the presence of TPEN reflects a weakening of the arrest potency of the AP caused by the abutting residues. It is unlikely that TPEN is responsible for the effect, as this would require some kind of highly specific interaction of TPEN and this precise construct. Since PURE has some residual Zn^2+^, it is necessary to use TPEN for the no-Zn^2+^ controls. This is now stated in the text.

3) The authors commented on the universality of the uL24 loop in bacteria. It would be great to also comment on the non-universality of uL23 or uL24 loops in other domains of life (like the absence of uL23 loop in spinach chloroplast ribosomes mentioned in the paper) and how it may suggest that co-translational protein folding may work differently in another organism due to differences in geometry of its ribosomal peptide exit tunnel.

We now include multiple-sequence alignments of uL23 and uL24 homologs as Figure 1—figure supplement 2, showing the absence/presence of the loops in different phyla (as a case in point, both the uL23 and uL24 loops are absent in mammalian ribosomes). We have also added a comment that the uL24 homolog in mammalian ribosomes does not obstruct the exit tunnel to the extent that it does in bacterial ribosomes, leaving a wider exit area. Indeed, in a recent paper (available on BioRxiv https://doi.org/10.1101/201913), we show that ADR1a folds at a somewhat shorter tether length in mammalian ribosomes than in *E. coli* ribosomes.

Reviewer 2:

[…] This study attempts to deconvolute the tunnel components that may be important for co-translational folding using the force assay which the authors are known for. This is a valuable, complex and rich experiment. It seems to me, however, not enough to interpret such complex events of translational and folding. This is my main concern: all conclusions are drawn based on the force experiments alone. Also, although some structural data are presented they are not used to rationalize the force results. It seems therefore the authors try to overstretch interpretation of their data.

We address the criticism that the conclusions are based on the force experiments alone in our response to reviewer 3 below. The structural data is included to show that the tunnel is indeed larger in size in uL23 Δloop ribosomes and that no other structural alterations are induced by the deletion, beyond the removal of the loop itself.

Concerning the cryo-EM, we have of course considered obtaining cryo-EM structures for each of the mutant ribosomes with folded domains in the tunnel. However, based on our previous studies where we have complemented the force-measurement assay with cryo-EM maps, we can conclude that there is a very good correlation between where in the tunnel protein domains fold as determined by force-profiles and as deduced from cryo-EM maps. Obtaining cryo-EM maps for ADR1a, spectrin, and I27 for each ribosome mutant is a massive project in itself, that at best would make it possible to be a little more precise in determining the location in the tunnel where folding is initiated but would not alter the main conclusion about the effects of tunnel size on folding.

As an alternative approach to cryo-EM, we now include coarse-grained MD simulations of the cotranslational folding of the ADR1a, R16, and I27 domains in WT, uL23 Δloop, and uL24 Δloop ribosomes, Figure 3 and Figure 3—figure supplement 10. *L_onset_* values (see below) are reproduced surprisingly well by these simulations, allowing us to visualize the location on the exit tunnel where these proteins start to fold, as shown in Figure 3—figure supplement 10B.

Overall the manuscript contains some nice experiments and this is an interesting study. The study however, seems to need more substance to support some of its more far reaching statements. For example, the broader implications at the end would be excellent, had the paper have more depth. For me, this study in its current form leaves too many open questions.

Indeed, as do most studies, the study raises new questions that will require substantial work to address; we nevertheless feel that it makes a clear and (hopefully) interesting point. This is why we submitted it as a Short Report.

1) Can the authors comment on the experimental conditions used for translation. Do the mutant ribosomes have the same translational kinetics as for wt? Are any differences likely to influence the force plots?

The ribosomes used have been tested for translational kinetics. Their translation rate in the PURE system used is between 0.4 – 0.5 amino acids/s, similar to published data (Capece et al., 2015). This is 10x slower than in vivo, but the proteins studied fold on micro-to-millisecond time scales, i.e., considerably faster than the in vivo translation rate. It thus seems safe to assume that the folding reaction equilibrates between each translation step both in vivo and in the PURE system. We have included a comment on this in the text.

2) The cryoEM structure of the uL23 ribosomes is interesting, but why the authors did not investigate programmed ribosomes as they have previously? However, using the uL23 structure, can the authors comment further on the structural environment that the ADR1a would have in the uL23 ribosome compared to WT ribosomes? Does it have more space than in the original location? Given the rationale that the authors use to describe where folding begins to takes place (e.g. ca. 17aa in uL23 deletion). More information regarding the significance of structure would help.

See comment above on the use of cryo-EM. Our data indicates that ADR1 starts to fold at L ≈ 17 residues in uL23 δ-loop ribosomes. Based on the MD simulations (above), we now provide more structural detail on how the deletion of the uL23 loop creates sufficient space for ADR1a to fold deeper in the tunnel.

3) It is not very clear to me how the authors define folding. Descriptions of the plots include words like "starting to increase, starting to fold". More information is needed on how these points are determined because to me it seems very subjective. In the absence of unfolded controls (e.g. in R15 and R16), the context for the "dips" and "rise/peaks" are difficult to rationalise.

We have clarified this point. To be precise, we extract three *L*-values from a given force profile: *L_onset_* (the *L*-value for which *f_FL_* has risen to its half-maximal value), *L_max_* (the *L*-value where *f_FL_* is maximal), and *L_end_*(the *L*-value where *f_FL_* has dropped to its half-maximal value). The interval *L_onset_* to *L_end_* corresponds to the tether lengths at which folding affects *f_FL_*. See comment below on validation of the force-measurement assay. Non-folding controls for both the spectrin domains and the I27 domain can be found in previous publications; we now refer to these publications in the legend of Figure 3.

4) On a similar point, overall, the analysis of Figure 3 appears ambiguous. For example, in Figure 3B red (uL23), data points from L=21-24 are effectively identical, all in error range, which casts potential doubt on the contention that there is a shift by 2-3 residues. The same types of difficulties arise in following analysis Figure 3D and 3E. One cannot clearly say there is 2-3 residue shifts. In order to focus on this small difference between the profiles, there should be more detailed analysis on other differences between the force profiles, for instance, why there are significant differences in fFL values between different ribosomes with the same length of nascent chains.

These shifts are now discussed based on the *L_onset_* and *L_max_* values presented in Table 1. We hope that this helps clarify our reasoning.

5) In ADR1a in the ΔuL23 ribosomes the first point (17aa) is non-zero even after subtracting the unfolded control. This suggests that the ADR1a is beginning to fold 51Ang (3ang per aa) from the PTC, approximately near uL22. If this is a folding, why is there never a clear peak? The authors say that the EM data tells us that the tunnel hasn't changed – even if the uL23 loop is gone and folding 'begins' early, (a) why isn't a complete fold reached (b) why would the ADR1a complete its fold in a narrow region of the tunnel rather than the vestibule exit? The "spring constant" idea is attractive – if the tunnel has opened up, then the NC can begin to sample conformations immediately, and in this case never reaches a maximum point of tension in the tunnel.

Concerning the *L*=17 point, see the response to reviewer 1. Concerning the tunnel in uL23 Δloop ribosomes, what we say is that the shape of the tunnel doesn’t change in any appreciable way beyond what is expected from the absence of the loop. Our main point is that the wider tunnel in the area normally occupied by the uL23 loop allows ADR1a to complete folding in this (no longer “narrow”) part of the tunnel. We now further illustrate this point with the MD simulations, Figure 3 and Figure 3—figure supplement 10.

6) There are differences in Figure 1B observed between wt ribosomes in the PURE-kit or those prepared n-house and no explanation is given as to why? The WT and the WT(purified) ribosomes also show different patterns, even if the peak appears to be retained. Can the authors comment on the general reproducibility? Additionally, the PURE-derived force profile has no error bars – were replicates performed? Why is Δf used solely for this graph?

We did not do replicates for the PURE-kit ribosomes because we were only interested in checking whether there were any obvious differences between the two preparations (there aren’t), not in making a precise comparison. All other results reported in the paper were obtained with the in-house preparation, and hence can be directly compared. Not to unnecessarily confuse the reader, we have removed the PURE-kit-ribosome data from the manuscript.

*7) The use of unfolded controls for ADR1a shows clearly need to subtract data – why was this not done for R15, R16 and I27 plots?*

A force-profile for a non-folding point mutant of I27 has already been published (Nilsson et al., 2017); it’s at background levels throughout and subtraction would make little difference. For the spectrin domains, *f_FL_* was measured for non-folding mutants at a few *L*-values close to *L_max_* and also shown to reduce *f_FL_* to near background (ref 13). We now refer to these data explicitly in the legend to Figure 3.

8) Overall, the inclusion of both R15 and R16 is perhaps somewhat strange: on one hand, analysis of R15 is more convincing that other nascent chains are used to elaborate on the differences in force profiles. However, can the authors elaborate further on the significance of R15 and R16 and how these results relate to their previous work? For the R15 force profile, the peak of the wild type does not look the same as the previously reported, as it has a peak of 37 amino acids. Although the authors explain that this is because a folding intermediate is formed this is not clear or supported by other data. They also attribute the observed difference in force amplitude between wt and ΔuL23 to the lower "spring constant" due to the wider tunnel in the latter. The discussion of a 'spring tension' was interesting, but not explored and it seemed a pity in this context. Again, supporting data are needed.

For R16, *L_onset_* and *L_end_* are clearly 2-3 residues smaller in uL24 Δloop ribosomes than in wt ribosomes; *L_max_* also appears shifted, but because *f_FL_* ≈ 1 around *L* = *L_max_*it is difficult to determine *L_max_* precisely in this case. Because of the generally low *f_FL_* values for R15 and the apparent “dip” at *L*=37 seen for WT ribosomes, it is difficult to unambiguously determine *L_onset_* and *L_max_*. We have abided by the reviewer’s advice and have removed the R15 data from the manuscript.

9) The folding intermediate proposed in R15 – why would this intermediate only exist in uL23 deleted and WT ribosomes, but not in uL24 deleted ribosome? This seemed to be unclear. If the protein off the ribosome is renatured from an unfolded state does it form intermediates on its folding pathway?

This is an observation that is somewhat tangential to the main point we’re trying to make in the paper. As said in above, we have removed the R15 data from the paper rather than making speculative suggestions concerning a possible intermediate.

10) It is not clear where in the ribosome the 6-9 Å measurement corresponds to in both the ΔuL23 ribosomes (subsection “ADR1a folds deeper in the exit tunnel in uL23 Δloop but not in uL24 Δloop ribosomes”, last paragraph) and in ΔuL24 ribosomes (subsection “Spectrin and titin domains fold deeper in the exit tunnel in uL24 Δloop ribosomes”, third paragraph). How does this correlate to the structure – i.e. is it consistent with the approximate location for the uL23 deletion?

The 6-9 Å calculation is based on assuming a largely extended conformation of the tether connecting the AP to the protein domains, i.e., ~3 Å per residue. We now further address the same point based on the new MD data.

11) The concluding statements appear to extrapolate from the data too much, and don't appear to be supported by the experimental data presented in the study – for example: "may prevent premature folding".…."that depend on interactions with TF…" Similarly, the uL24 loop statements about correlating length and sequence is not particularly well supported either – it seems to be a rather loose conclusion to an otherwise interesting paper.

Obviously, these are speculative ideas that we bring up in order to delineate questions for further study. We have removed this speculation.

Reviewer 3:

[…] My major concern with this manuscript is that the data presented do not adequately support the conclusions. The only experimental technique used is an assay in which of a SecM translational arrest peptide acts, broadly speaking, as a force sensor, whereby the measurements are performed by SDS-PAGE.It is very difficult to assess the errors associated with this assay, in part because the primary data are not reported.

We now provide Figure 3—figure supplements 1-7 with gel images, and an excel sheet with source data.

Most importantly, however, these conclusions should be validated using an independent technique. In the absence of this validation, one should put an excessive trust in the assay used, which would seem to be low resolution at best.

We realize that we should have been more explicit in summarizing previously published data that validate the force-measurement technique in studies of cotranslational folding. In short, (i) folded proteins have been visualized in the ribosome tunnel by cryo-EM of RNCs at *L*-values close to *L_max_* (ADR1a, spectrin R16) and *L_onset_*(the I27 domain), (ii) the ± Zn^2+^ data for ADR1a shows that the *f_FL_* peak is caused by folding into the native state; this is further supported by an Ala-scan of the entire ADR1a domain showing that the only mutations (except for the Zn-binding residues) that affect *f_FL_* at *L* = *L_max_* are Leu(18) to Ala and Phe(12) to Ala (Samelson et al., 2018), the two residues that define the hydrophobic core of the protein, (iii) *L_onset_* has been shown to correspond closely to the tether length at which protein domains become resistant to on-ribosome pulse-proteolysis by thermolysin (Farias-Rico et al., 2018 https://doi.org/10.1101/303784) for proteins SOD1, ILBP, and S6, (iv) peaks in the *f_FL_* profiles of calmodulin and ILBP disappear when translation is done in the presence of EGTA or absence of ligand, respectively, and when hydrophobic residues in the core are mutated to Ala in 3 other proteins (Farias-Rico et al., 2018), (v) the *f_FL_* profile for I27 is quantitively reproduced by a physical model of the force-measurement assay based on molecular dynamics simulations (Nilsson et al., 2017); preliminary data show that this also holds for the I27 profile obtained with uL24 Δloop ribosomes. We now cite these papers en bloc in support of our interpretation of the *f_FL_* profiles. If deemed necessary, we’ll be happy to expand the discussion of this point, but the manuscript is already on the long side for a Short Report.

We feel that the force-measurement assay has been quite extensively validated already, on par with other assays for cotranslational folding such as Rodnina’s FRET assay (Holtkamp et al., 2015) or Christodolou’s NMR assay (Cabrita et al., 2016); obviously, each assay has its own strengths and weaknesses. In contrast the reviewer 3, we think that the force-measurement assay is of high resolution: standard errors are small, and peaks in the *f_FL_* profiles can be mapped at single-residue resolution. We do think, however, that more information may be extracted from the *f_FL_* profiles than we do at present (especially from the *L_end_* values); this will require further work, but does not affect the conclusion about deeper folding in wider exit tunnels made in the present study.

[Editors’ note: the author responses to the re-review follow.]

[…] Before publication, we encourage you to include in the paper more context for the importance and significance of the work, such as provided in your recent PNAS paper (Farias-Rico et al., 2018). We also suggest that you include some additional explanation for the validation of the method, as you suggest in your response to reviewers. You may do this as you think best suits your purposes, and we will approve your revised manuscript for publication without further delay.

We have done two things in the final revision:

1) We have spelt out the different ways that our folding assay has been validated against other methods at the end of the section “The Folding Assay”.

2) We have added a final sentence at the very end of the main text to put the work in an evolutionary perspective.